# Individual and contextual factors affect the implementation fidelity of youth-friendly services, northwest Ethiopia: A multilevel analysis

Yohannes Ayanaw Habitu[1]*, Gashaw Andargie Biks[2], Abebaw Gebeyehu Worku[1], Kassahun Alemu Gelaye[3]

1 Department of Reproductive Health, Institute of Public Health, College of Medicine and Health Sciences, University of Gondar, Gondar, Ethiopia, 2 Department of Health System and Policy, Institute of Public Health, College of Medicine and Health Sciences, University of Gondar, Gondar, Ethiopia, 3 Department of Epidemiology and Biostatistics, Institute of Public Health, College of Medicine and Health Sciences, University of Gondar, Gondar, Ethiopia

* yohaneshabitu@gmail.com

**Data Availability Statement:** All relevant data are within the paper.

**Funding:** This study is part of a PhD dissertation and received money from University of Gondar for

## Abstract

### Background

The evaluation of all potential determinants of implementation fidelity of Youth-Friendly Services (YFS) is crucial for Ethiopia. Previous studies overlooked investigating the determinants at different levels. Therefore, this study aimed to assess the determinants of implementation fidelity of YFS considering individual and contextual levels.

### Methods

This study was conducted among 1,029 youths, from 11 health centers that are implementing the YFS in Central Gondar Zone. Data were collected by face to face interview and facility observation using a semi-structured questionnaire. A Bivariable multi-level mixed effect modelling was employed to assess the main determinants. Four separate models were fitted to reach the full model. The fitness of the model was assessed using Akaike Information Criterion (AIC) and level of significance was declared at p-values < 0.05. The results of fixed effects were presented as adjusted odds ratio (AOR) at their 95% CI.

### Results

Four hundred one (39.0%) of the respondents got the YFS with high level of fidelity. Had high level of involvement in the YFS provision (AOR = 1.35, 95% CI: 1.15, 1.57), knew any peer educator trained in YFS (AOR = 1.60, 95% CI: 1.36, 1.86), and involved as a peer educator (AOR = 1.46, 95% CI: 1.24, 1.71), were the individual level determinants. Whereas, got capacity building training; (AOR = 1.93, 95% CI (1.12, 3.48), got supportive supervision, (AOR 2.85, 95% CI (1.99, 6.37), had a separate waiting room (AOR = 9.84, 95%CI: 2.14, 17.79), and system in place to provide continuous support to staff (AOR = 2.81, 95%CI: 1.25, 6.34) were the contextual level determinants.

data collection only. The funder had no role in study design, data collection and analysis, decision to publish, or preparation of the manuscript. The authors declare that they have received no funds for publication of this manuscript and they have no external source of fund for both data collection and publication.

**Competing interests:** The authors have declared that no competing interests exist.

## Conclusions

The level of implementation fidelity remains low. Both individual and contextual level determinants affect the implementation fidelity of YFS. Therefore, policy makers, planners, managers and YFS providers could consider both individual and contextual factors to improve the implementation fidelity.

## Background

Youth-Friendly Services (YFS) are Evidence Based Practices (EBPs) that are available, accessible, acceptable, appropriate, and equitable for youth [1, 2]. The YFS intervention was developed by World Health Organization (WHO) with the intent to avert the Sexual and Reproductive Health (SRH) problems among youth [1–3]. The intervention has previously been highlighted as a successful model for providing SRH services within a public health system [1, 4]. In Ethiopia, by 2012, YFS was implemented and provided with integration to the public health system [5].

The YFS is a complex intervention having many components integrated and provided in a room [4]. A range of services like, counseling and provision of information on SRH, Human Immunodeficiency Virus (HIV) testing, gynecological examinations, pregnancy testing, contraceptive provision, management for Sexually Transmitted Infections(STIs), abortion care, and more are integrated in YFS [1, 4]. Implementation of complex programs like YFS in real-world setting involves multifaceted processes, which are often influenced by several levels of determinants [6], hence implementation rarely succeeded as intended [6–8].

Even if YFS is widely implemented, youth in developing countries(including Ethiopia), suffer from many SRH diseases like STIs including HIV/AIDS, unsafe abortion, unintended pregnancy and child birth and more [3, 9]. One of the possible reasons for the high prevalence of SRH problems among youth could be the YFS intervention may not be implemented as designed by the original program developers. Little evidences showed as the YFS are not implemented as intended [10, 11].

Program implementation is influenced by contextual factors, which impact on an organization's capacity to implement with high fidelity [12–16]. Hasson, operationalized context as factors related to levels of policies, finances, organizations, and groups of participants [17]. Contextual factors like providers training and competency [18–20], supportive context and skilled providers, and receptive participants [21] are some of the determinants of program implementation with fidelity. Other studies verified pre-implementation training, presence of detailed delivery manuals or guidelines, ongoing support or supervision as main determinants of program implementation [12–14, 19, 22, 23].

Literature on similar interventions showed contextual factors like, lack of technical support and monitoring, limited resources, providers commitment [24–26], poorly designed facilities [27], inconvenient opening hours [28], long waiting times and distances to the health centers [29] as the main determinants of fidelity of implementation (FoI) of interventions. In addition, operating hours, travel time, costs [4], program specific training [27], poor providers counseling skills [30], year of experience [6, 12], provider competency, as the main determinants of FoI of interventions [7, 12–14, 17, 22].

Evidence from other interventions showed, provider characteristics like provider's perception of the program [12, 14], provider's attitude towards an intervention and their motivation to fully implement the program [14, 15], provider's judgments about, and confidence in,

having the skills required for implementing the program [12, 14, 15], are some of the determinants of implementation fidelity of interventions. Moreover, participant responsiveness is one of the factors affecting FoI of interventions at an individual level [7, 13, 18, 31, 32]. Investigating the determinants of FoI of YFS using the multilevel perspective is very important, in that, it will uncover evidences related to the determinants of implementation fidelity of YFS. In addition, the finding from this research could help health care providers, planners, programmers and decision makers working on YFS to have a good insight on determinants of FoI of YFS at each level, to take appropriate measures and this in turn may strengthen the provision of YFS with fidelity, and to reach the desired intervention outcomes. Therefore, the aim of this study was to determine the determinants of FoI of YFS using the multilevel modeling approach.

## Materials and methods

### Design and context

A cross-sectional study with a linked YFS program and individual youth survey was conducted, from September to December of 2019 in Central Gondar Zone. These surveys were conducted concurrently to analyze all potential determinants of implementation fidelity of YFS at two levels YFS program (level-two) and youth level determinants (level-one). The use of linked survey is usually considered as stronger approach to analyze causality in non-experimental studies like this [33].

### Participants and setting

The source populations of the study were youth of age 15–24 who were using YFS in the study area and the study population were youth of age 15–24 who were using YFS in that specific area and available at the data collection period. Youth, who provided informed written consent, parental consent or assents during the data collection period, were included in the study. In addition, health centers providing YFS, the YFS program, and health care professionals working on YFS were also another study population. Health centers and the YFS program were assessed for their level of readiness to provide the YFS using a standard checklist. In addition, the YFS providers were assessed to show the providers competency in the YFS provision.

Central Gondar Zone has a total of 14 districts with 430 kebeles (the smallest administrative units locally), 76 public health centers (from which 35 health centers were implementing YFS). The total population resided in the aforementioned zone was, 2,265,200. Of these, 1, 1411,325 were males according to the 2018 Zonal report [34]. There were 807,606 youth aged 10–24 years in the zone accounting for 36% of the total population. Besides, Gondar city administration (the capital city of Central Gondar Zone) is located at the center of this zone and had 25 urban and 11 rural kebeles [34]. According to the city 2018 plan, Gondar City Administration had a total of 390,644 populations, and youth population accounted 111,325. In addition, the city had also a total of 8 health centers and all the health centers were implementing the YFS since [34].

### Description of the intervention (YFS)

The YFS intervention was developed by WHO by the year 2000 [1, 2]. It was designed to address youth SRH service demands and hence to avert the impact of SRH problems among youth. The YFS intervention implementation strategy was designed to be delivered as an integrated service with in the public health system in a one-stop-shop approach. A range of SRH and other health care services (around 11 services) are integrated and provided for youth in a

room designed for this intervention. In addition, in a single visit youth can get a range of services starting from counseling on SRH issues to other SRH and/ medical conditions. Moreover, health care providers first get pre-service training on the provision of the YFS intervention.

## Sample size determination

To determine the sample size for this study, a pilot study was conducted among 60 youth residing in similar setting but was not included in the final study. One health center from Central Gondar Zone (Enfranz health center), and two health centers in Bahir Dar City (Han and Bahir Dar health center) were included in the pilot study. These health centers had already implementing YFS. Then the single population proportion formula [35] was used considering the assumptions of the proportion of high level of fidelity, 26.7%, from the pilot study, and considering 95% CI, margin of error of 4%, design effect of 2, and 10% non-response rate. The final calculated sample size was 1,034. In addition, 11 health centers providing YFS and 27 YFS providers working in those 11 health centers were included in the study.

## Sampling procedure

In Central Gondar Zone, there are 14 rural districts and one city administration. Out of the 14 rural districts, 5 districts were selected by simple random sampling technique. Then, if there are two or more health centers implementing YFS in each district, 1 health center, was chosen by random sampling. Hence, 5 health centers from the Central Gondar Zone (Amba Giorgis, Maksegnit, Kolladiba, Chuahit and Delgie) were selected and included. On the other hand, from the 8 health centers that were implementing the YFS in Gondar city administration, 6 health centers (Gondar, Azezo, Tseda, Gebriel, Woleka and Maraki) were selected randomly and included in the study. Finally, when we sum up those selected health centers (from the Central Gondar Zone (5 HCs) and Gondar City Administration (6 HCS)), a total of 11 health centers were included and considered as clusters. Within each cluster the YFS program strength, the health facility readiness and the YFS providers' competency was assessed.

Then the sample size was proportionally allocated according to the size of the population in each health center to get representative participants in each selected health center. Finally, youths were selected by systematic random sampling technique, in all working hours of the week during the data collection period in each health center.

## Data collection instrument and procedures

**Data collection instrument.** The instrument which has 17 general items; which measured the socio-demographic and other individual characteristics was used. In addition, a validated tool, comprised of 65 items with 5category Likert scaled items was designed to measure the fidelity of implementation of YFS. Moreover, an instrument having 73 items was used to observe and evaluate the YFS program competency. Furthermore, another instrument having 38 items was used to assess the YFS providers' competency and lastly an instrument having 90 items was used to assess the health facility readiness. All the tools used to assess the YFS program strength, the YFS providers' competency and the health facility readiness were adopted from the WHO standard [36].

Facility level factors are those characteristics like if the health facility has signal listing for all the YFS available or not, if the health facility had a separate discreet entrance for youth to ensure youth privacy or not, if the health facility offered YFS for free or at rates affordable to youth, if the health facility have adequate fund allocated for YFS, if the health facility had clear, written guidelines or standard operating procedures exist for YFS and the like.

To Measure the implementation fidelity of services like YFS are conceptually developed from three major constructs called adherence, quality of service delivery and participant responsiveness [37, 38]. Hence, it is vital to quantify the three main constructs that are intended to measure fidelity of YFS [37, 38]. To measure the overall fidelity of YFS, the three constructs used to measure the fidelity of YFS (adherence, quality of YFS delivery and participant responsiveness) were quantified separately. Then the overall fidelity score was computed. The 65 items scales were mainly from the WHO-Plus standard tool (quality assessment tool) [36]. The scale was developed to measure the three dimensions of fidelity i.e. adherence, quality of service delivery, and participant responsiveness. Then a fidelity score was developed for each fidelity domain, based on 5 Likert scale level. The scale passed the standard tool validation process, starting from face validity, content validity, construct validity, pilot tested and finally internal consistency was checked using the information from the pilot study. Finally, 9 items were developed to measure the participant responsiveness dimension, with high internal consistency (Cronbach's alpha value of 0.85), 15 items were developed to measure the adherence dimension with Cronbach's alpha value of 0.91 and 41 items were developed to measure the quality of delivery dimension, with high internal consistency having Cronbach's alpha value of 0.93. The quality of delivery dimension was further constructed based on the Donavidian model quality of care framework, that aimed to assess the structural quality (11 items), process quality (23 items) and outcome quality dimensions (7 items).

Some of the questions used in the fidelity measure based on the three domains (adherence, quality of delivery and participant responsiveness) are described below. Questions used in the adherence domain were: Confidentiality of the service was assured for you, provider was respectful to you and the provider explained to you on all the range of available YFS there. Questions used in the quality of delivery domain were: The hours and day that you came to the facility were convenient for you, you were very clear of the information given by the provider, you are welcomed and get the YFS without appointment, and the provider encouraged you to ask any questions. Questions used in the participant responsiveness domain were: You were involved as a peer educator in YFS, you were involved in contributing to decisions about how health services should be delivered to youth clients, and you were involved in YFS service design and delivery.

In this study level of youth engagement or participant responsiveness is defined as the participation of youth in the YFS intervention in aspects like participation in the YFS design, planning and delivery, participation in the YFS as a peer educator (counselor), involvement in the YFS on decisions about how health services should be delivered to youth clients and the like.

Data were collected by an interviewer administered, predetermined and structured questionnaire. Eleven BSc holders (5 Health Officers and 6 Midwives), who had special training on YFS and working out of the data collection area, collected the data. The data collectors were not involved in the implementation of YFS in the study area. One supervisor having a master of public health and with work experience on supervision in research data collection was involved. In addition, structured interview and direct observation of the health centers were used to collect data regarding the YFS provider competency, health facility readiness and YFS program competency.

**Variables of the study.** In this study, the outcome variable was the fidelity of implementation of YFS, which was dichotomized in to high fidelity groups and low fidelity groups. The authors' of this study were interested to investigate the effects of variables at two levels on the fidelity of implementation of YFS.

**Level 1 variables:** factors related to individual youth who utilized the YFS like; had high level of involvement in the YFS provision, knew peer educator trained on YFS, involved as a

peer educator, involved in YFS service design and delivery, and level of involvement in making decision regarding your treatment was very high.

**Level 2 variables:** YFS program level variables were considered as level 2/context level factors which are described below. Variables like presence or absence of health facility on; signal listing YFS available, counseling area that ensured visual privacy, examination room that ensured auditory privacy, separate discreet entrance for youth to ensure their privacy, separate waiting room for youth clients, peer education program, educational posters displayed, and services provision was attractive and friendly to youth. In addition, variables like YFS provider's age, sex, educational level, got pre-service training or not, means of capacity building training were provided, assessed using quality standard checklists, all staff were oriented to provide confidential YFS, youth clients' privacy and confidentiality were ensured, used visual materials to help you in your daily work, and used computers to help you in your daily work were considered. Furthermore, variables like presence or absence of support and commitments that the RHBs have made towards YFS, human resource allocation adequate in terms of the volume of work, youth involved in monitoring the quality YFS, written guidelines for staff, system in place to provide continuous support to staff, policies and strategies that help youth to be involved in decision-making, written guidelines or standard operating procedures exist, the program publicize the services available to youth by stressing confidentiality, staff or volunteers who do outreach activities, and more.

## Operational definitions

Fidelity of implementation: is defined as the extent to which youth get the YFS intervention as compared to the original YFS program protocol based on the three domains called adherence, quality of service delivery and participant responsiveness.

High fidelity of implementation groups: In this study, those youth who receive the YFS with total fidelity score of greater than or equal to 60%($> = 195/325$) [7].

Low fidelity of implementation groups: Those youth who receive the YFS with total fidelity score below 60% ($<195/325$).

## Data quality control

To control the data quality, three days training was provided to 11 data collectors and a supervisor before the actual data collection period. In addition, the instrument was validated (face validity, content validity, construct validity, internal consistency was high). Moreover, appropriate modifications were made on the instrument, after conducting the pilot study. The questionnaire was first translated in to the local (Amharic language) by a language and a professional experts. Furthermore, it was back translated to English language by another one language expert and one professional expert. Then, to ensure consistency, the Amharic version of the instrument was back translated in to English language by another English language expert and by another professional expert.

## Statistical analysis

The collected data were manually cheeked for completeness, entered in to EpiData software version 3 and exported to STATA version 14 for further analysis. First, descriptive analyses summaries, frequency, and percentages were done on the characteristics of the study population the explanatory variables, each fidelity construct and overall fidelity score of implementations of YFS presented in terms of frequency, percentages and tables.

The unit of analysis was at an individual level and then aggregated in to the health center level in order to compare the level of fidelity of implementation in all health centers. Then

individuals and facilities with total implementation fidelity scores were graded separately in to two levels. In this study, those youth whose total fidelity score of greater than or equal to 60% ($> = 195/325$) [7], were declared as had received the YFS intervention with high FoI, and those youth whose total fidelity score below 60%($<195/325$) were declared as had received the YFS intervention with low FoI. In this study, the WHO cutoff value ($> = 75\%$, $\geq243.75/325$) was not used in the data analysis and interpretation sections to declare good fidelity. The reason why we did not used the WHO cutoff value was, while we were using the WHO cutoff value, the proportion of youth who get the YFS with higher fidelity became very small, 48 (4.7%). Which made the data analysis very difficult and therefore running/fitting/ the multi-level modeling using such small proportions was impossible. Hence, we reviewed the available evidence and used 60% as a cut off value. Finally, the individual level and health facility level data sets were merged and linked for analysis using the STATA merge command.

A two-level multivariable multilevel logistic regression analysis was applied with fitting four different models. The rationales for using a multilevel modeling were the following. Firstly, the FoI patterns YFS are influenced by the characteristics of different levels (individuals and contextual factors like YFS program, YFS providers and health facilities). Analyzing variables from different levels at one single common level using the standard binary logistic regression model leads to bias (loss of power or Type I error) [39, 40]. This approach also suffers from a problem of analysis at the inappropriate level (atomistic or ecological fallacy). Multilevel models allow us to consider the individual level and the group level in the same analysis, rather than having to choose one or the other. Secondly, due to the multistage cluster sampling procedure, individual youth were nested within health centers; hence, the likelihood of youth seeking YFS is likely to be correlated to the health care providers, facilities, accessibility and availability. The assumption of independence among individuals within the same cluster and the assumption of equal variance across clusters are violated in the case of nested data. Hence, the multilevel analysis is the appropriate method for such cases [39, 40].

In this study, the following equation elaborates the multilevel analysis for FoI of YFS, the link function is logit, and we get the logistic regression model as:

$$Log[\pi_{ij}/(1 - \pi_{ij})] = \beta_0 + \beta_1 X_{1ij} + \ldots + \beta_n X_{nij} + u_{Oj} + e_{ij}$$

Where, $\pi_{ij}$ is probability of the presence of High FoI of YFS, $(1- \pi_{ij})$ is probability low FoI of YFS, $\beta_0$ is log odds of the intercept, $\beta_1 \ldots \beta_n$ are effect sizes of individual and health center level factors, $X_{1ij} \ldots X_{nij}$ are independent variables of individuals and HCs, $u_{Oj}$ are random errors at cluster level, and $e_{ij}$ show random errors at individual level. The distribution of $u_{0j}$ is normal with mean 0 and variance $\sigma^2 u_0$, the random effect was explained using ICC, which was calculated using between-cluster variance and within-cluster variance [ICC = $\delta^2 u_0/\delta^2 u_{0+}\pi^2/3$], in log distribution, the residual variance of FoI of YFS within a cluster is zero but variance is considered constant at $\pi^2/3$(where, $\pi^2/3$ denotes the variation within a cluster and $\delta^2 u_0$ is the variation between clusters. The ICC was used to show the level of between-cluster variation and finally we used the Variance Inflator Factor (VIF) to examine instability of effect size of predictors as the result of high collinearity among the factors.

## Steps in multi-level modelling

Screening for determinants at level one and level two for FoI of YFS was done by conducting bivariable logistic regression analysis separately. Then factors having p-value of $<0.2$ in the Bivariable model were selected and fitted in to the Multi-level modeling [41]. Then four separate models were fitted to reach the full model. First the null model (model I), contained no exposure variables was run which was used to test the random effect of between and within-

cluster variability by determining the Intra-Cluster Correlation (ICC). Then model II that was adjusted for individual-level variables, with the fixed level one determinant with randomly varying intercepts. The effects of individual level characteristics on FoI of YFS were determined. Next, Model III was adjusted for level two determinants with randomly varying intercepts. Finally, model IV (full model), adjusted for both individual and contextual level variables, and with fixed level 1 and level 2 predictors with randomly varying intercepts and slope were fitted. The important characteristics of individual youth and clusters were concurrently fitted to one model to reveal their net fixed and random effects.

Statistically significance association was declared using two-tailed test and at p-values less than 0.05. The results of fixed effects were presented as adjusted odds ratio (AOR) at their 95% confidence interval (95% CI) after considering potential confounders. Random effects were expressed in terms of Intra Class Correlation Coefficient (ICC) that explains the amount of health center variation. The clustered nature of the data, and the within and between health center variations were taken in to account by assuming each health canter has different intercept ($\beta_0$) and fixed coefficient ($\beta$). Proportional Change in Variance (PCV), expresses the change in the community level variance between Model-I (empty model) and the consecutive models (Model-II, III and IV) [42].

## Model fitness and precision

The fitness of the model was assessed using Akaike Information Criterion (AIC), AIC was used to choose a model that best explains the data and the model with low AIC value was taken [43]. A test of how well the model explains the data (goodness of fit test) was checked by using Hosmer-Lemshow statistics and it was non-significant (prob> chi2 = 0.1270), indicating the model fits the data reasonably well. The multicollinearity (correlation of predictors with each other) was checked by using variance inflation factors (VIF) and no variable had VIF greater than 10 as a cut off value, indicated the absence of significant collinearity among explanatory variables [44]. Two-tailed Wald test at significance level of alpha equal to 5% was used to determine the statistical significance of the determinants and all the analyses were performed with Stata SE 14 software package.

Five variables from the individual level variables, and 19 variables from the higher level variables that fulfilled the screening criteria were selected and fitted in to the multi-level modeling. In summary, a total of 24 variables from individual and contextual level factors were fitted in to the multi-level model.

## Ethical considerations

Ethical clearance was obtained from the University of Gondar, Institutional Review Board (IRB) with reference number, R. no.-O/V/P/RCS/05/1047/2019. Official permission was obtained from respective Zonal and local authorities to cascade data collection. Informed and written consent was sought from each study participant. In addition, for those respondents of age below 18 years individual assent and parental consent was obtained. Moreover, confidentiality was maintained through anonymity and privacy measures to protect respondent's right through the research process. Moreover, respondents were informed about their right to withdraw from the study at any time and they could not be harmed by doing so.

## Results

### Socio-demographic and other characteristics of the youth

Of the total 1034 youths, 1,029 (99.5%) responded to the survey. The majority of the respondents 717(69.7%) were aged between 20–24 years, while 752(73.1%) were females. Regarding

their religion, 874(84.9%), were Orthodox Christians, while 781(75.9%) were urban residents. Concerning their educational status, 453(44.0%) of them were attending secondary education (grade 9–12), and 601(58.4%) were not married (**Table 1**).

## Characteristics of the YFS providers

A total of 27 health care providers working on YFS in the 11 health centers were included in the study. Males accounted 15(55.6%), while 22(81.5%) had pre-service training on YFS. Regarding their profession, 12(44.4%) were clinical nurses at diploma level, 13(48.2%) were BSc nurses and 2 (7.4%) were BSc health officers. Twenty (74.1%) of the services providers used the National Adolescents and Youth Reproductive Health Services Strategy as a reference, while 8(29.6%) were supervised by higher officials, who were using the national YFS quality standard checklist **(Table 2).**

## The level of implementation fidelity and respondents' level of engagement on Youth-Friendly Services intervention

The results of the fidelity of implementation of YFS showed that 401(39.0%) of youths got the YFS with high level of FoI. Four hundred fifty five (44.2%) of the respondents were involved in

**Table 1. Socio-demographic and other characteristics of youth Northwest Ethiopia in 2019.**

| Variables | Frequency | (Percent) |
|---|---|---|
| **Age (in years)** | | |
| 15–16 | 58 | 5.6 |
| 17–19 | 254 | 24.7 |
| 20–24 | 717 | 69.7 |
| **Religion** | | |
| Muslim | 147 | 14.3 |
| Orthodox | 874 | 84.9 |
| Others* | 8 | 0.8 |
| **Educational status** | | |
| Unable to read and write | 58 | 5.6 |
| Able to read and write | 5 | 0.5 |
| Primary education (1–8) | 210 | 20.5 |
| Secondary education (9–12) | 453 | 44.0 |
| Vocational/Diploma | 211 | 20.5 |
| Degree and above | 92 | 8.9 |
| **Work for money** | | |
| No | 593 | 57.6 |
| Yes | 436 | 42.4 |
| **Mother alive at the time of the survey** | | |
| No | 126 | 12.2 |
| Yes | 903 | 87.8 |
| **Father alive at the time of the survey** | | |
| No | 312 | 30.3 |
| Yes | 717 | 69.9 |
| **Do you have peer friend/s at the time of the survey** | | |
| No | 201 | 19.5 |
| Yes | 828 | 80.5 |

Others implied

*protestant and Catholic

**Table 2. Characteristics of the YFS providers, Northwest Ethiopia, 2019.**

| Variables | Frequency | (Percent) |
|---|---|---|
| **YFS providers educational level** | | |
| Degree | 18 | 66.7 |
| Diploma | 9 | 33.3 |
| **Trained on YFS** | | |
| Yes | 22 | 81.5 |
| No | 5 | 18.5 |
| **There were means of capacity building training provided to you** | | |
| Yes | 7 | 25.93 |
| No | 20 | 74.07 |
| **You have got supportive supervision** | | |
| Yes | 8 | 29.63 |
| No | 19 | 70.37 |

decision-making regarding the YFS, while 341(33.3%) had high overall level of engagement in the provision of YFS. Two hundred eighty six (27.8%) of them were involved as a peer educator in the YFS, and 212(20.6%) were involved in YFS service design and delivery **(Table 3)**.

## Health facility and Youth-Friendly Services program level characteristics

Five out of the eleven health centers had signal listing YFS available, while 6/11 had a separate discreet entrance for youth to ensure their privacy. Six out of the health centers had counseling area that provided for visual privacy, while 7/11 offered YFS for free or at affordable rates to youth. Three out of the eleven health centers had peer education program available, while 8/11 had educational posters displayed in the health center **(Table 4)**.

Regarding the YFS program level characteristics, 4/11 health centers got support from the regional health bureau, while 5/11 had adequate fund allocated for the YFS. Eight out of eleven health centers had written guidelines for the staff (who were providing the YFS in the HC), while 7/11 of the HCs had system in place to provide continuous support to staff who work on the YFS. Eight out of the eleven HCs had clearly written guidelines or standard operating procedures (SOPs) in the YFS room, while 5/11 HCs had private registration process **(Table 4)**.

**Table 3. Youths level of engagement in the YFS intervention, Northwest Ethiopia, 2019.**

| Items | Responses | |
|---|---|---|
| | Yes | No |
| | Frequency (%) | Frequency (%) |
| You were involved in decision-making regarding the YFS | 455(44.2) | 574(55.8) |
| Your overall level of involvement in the provision of YFS was high | 341(33.3) | 688(66.7) |
| You have been involved as a peer educator in YFS | 286(27.8) | 743(72.2) |
| You know any peer educator trained in YFS | 337(32.7) | 692(67.3) |
| You are involved in decisions about how YFS should be delivered to youth | 283(27.5) | 746(72.5) |
| You are aware of youth who are involved in decisions about how YFS should be delivered to youth | 283(27.5) | 746(72.5) |
| level of involvement in making decision regarding your treatment was very high | 407(39.5) | 662(60.5) |
| Community involvement in YFS program design, monitoring and evaluation was high | 229(22.2) | 800(77.8) |
| Involvements in YFS service design and delivery were high | 212(20.6) | 817(79.4) |

**Table 4. The health facility and YFS program level characteristics on YFS, Northwest Ethiopia, 2019.**

| Characteristics | Responses |
|---|---|
| | **Frequency** |
| The counseling area kept visual privacy | |
| Yes | 6/11 |
| No | 5/11 |
| The examination room kept auditory privacy | |
| Yes | 6/11 |
| No | 5/11 |
| There was a separate entrance to ensure youth privacy | |
| Yes | 6/11 |
| There was a transparent and confidential system to submit youths' comments | |
| Yes | 4/11 |
| No | 7/11 |
| There was a separate waiting room for youth | |
| Yes | 4/11 |
| No | 7/11 |
| There was adequate waiting room for youth | |
| Yes | 5/11 |
| No | 6/11 |
| There were educational posters displayed | |
| Yes | 8/11 |
| No | 3/11 |
| There were posters that describe clients' rights | |
| Yes | 8/11 |
| No | 3/11 |
| There were materials for youth clients to take home | |
| Yes | 8/11 |
| No | 3/11 |
| Service provision was attractive and friendly to youth | |
| Yes | 7/11 |
| No | 4/11 |
| System in place to provide continuous support to YFS staff | |
| Yes | 7/11 |
| No | 4/11 |
| Clear, written guidelines or SOPs exist in the YFS | |
| Yes | 8/11 |
| No | 3/11 |

## Determinants of implementation fidelity of Youth-Friendly Services

The multilevel analysis was started by the intercept only model, to test the null hypothesis, that stated there is no variation in FoI of YFS between clusters (HCs) and to decide in evaluation of the random effects at the health facility level. The results presented in Table 4 indicated that considerable heterogeneity between health facilities was observed for each indicator of FoI of YFS. In all the three indicators, FoI of YFS was clustered significantly by HC. The intra-class correlation in the empty model for FoI of YFS indicated that 16.4% of the total variance in FoI of YFS was attributable to the differences across HCs (**Table 5**).

Table 5. Parameter coefficients and model comparisons of each successive model in FoI of YFS, Central Gondar Zone, 2019.

| Random effect | Model-I | Model-II | Model-III | Model-IV |
|---|---|---|---|---|
| Community variance (SE) | 0.63655 | 0.3396243 | 5.11e-32 | 1.22e-33 |
| ICC (%) | 16.4% | 23.5% | 2.1% | 2.4% |
| PCV (%) | Ref | 99.9% | 1% | 1% |
| Model comparison statistics | | | | |
| Log likelihood | -630.22393 | -478.29331 | -476.153 | -431.60187 |
| AIC | 1264.448 | 1084.603 | 877.1524 | 817.8602 |

## Individual level effects

The final Multi level modeling analysis result showed that from the individual level factors, factors like youth who had high overall level of involvement in the provision of YFS (AOR = 1.35, 95%CI: 1.15, 1.57), youth who knew any peer educator there trained in YFS (AOR = 1.60, 95% CI: 1.36, 1.86), and youth who were involved as a peer educator in YFS (AOR = 1.46, 95% CI:1.24, 1.71), were statistically significant determinants of the FoI of YFS. The odds of getting the YFS with fidelity was nearly 1.4 times higher among youth who had high overall level of involvement in the provision of YFS as compared to those who had not with (AOR = 1.35, 95%CI:1.15, 1.57). The odds of getting the YFS with fidelity was 1.6 times higher among those youth who knew any peer educator there trained in YFS as compared to those who did not know with AOR 1.60, 95%CI(1.36, 1.86). The odds of getting the YFS with fidelity was nearly 1.5 times higher among youth who have been involved as a peer educator in YFS as compared to those youth who were not involved as peer educator with AOR 1.46, 95% CI(1.24, 1.71) (Table 6).

Table 6. Bivariable and multivariable multi level logistic regression analysis of individual and contextual determinants of implementation fidelity of YFS, Northwest Ethiopia, 2019.

| Fixed effects of individual and contextual level variables | | Model-I | Model-II | Model-III | Model-IV |
|---|---|---|---|---|---|
| | | | AOR [95%CI] | AOR [95%CI] | AOR [95%CI] |
| Individual level determinants | | | | | |
| Your overall level of involvement in the YFS provision was high | Yes | - | 1.41(1.21, 1.63) | - | 1.35(1.15, 1.57)* |
| | No | - | 1 | - | 1 |
| You know any peer educator trained in YFS | Yes | - | 1.83(1.58, 2.11) | - | 1.60(1.36, 1.86)* |
| | No | - | 1 | - | 1 |
| You have been involved as a peer educator | Yes | - | 1.47(1.26, 1.72) | - | 1.46(1.24, 1.71)* |
| | No | - | 1 | - | 1 |
| Program level determinants | | | | | |
| There is a separate waiting room for youth | Yes | - | - | 7.85(5.28, 9.06) | 9.84(2.14,17.79)* |
| | No | - | - | 1 | 1 |
| There are means of capacity building training provided to you | Yes | - | - | 3.40(2.51, 5.72) | 1.93(1.12, 3.48)* |
| | No | - | - | 1 | 1 |
| You have got supportive supervision | Yes | - | - | 1.71(1.28, 4.02) | 2.85 (1.99, 6.37)* |
| | No | - | - | 1 | 1 |
| System in place to provide continuous support to YFS staff | Yes | - | - | 4.92(2.63, 7.04) | 2.81(1.25, 6.34)* |
| | No | - | - | 1 | 1 |

*P-value <0.05

### Contextual-level effects

Factors like health care providers who got capacity building training AOR 1.93, 95% CI(1.12, 3.48), health care providers who got supportive supervision, AOR 2.85, 95% CI (1.99, 6.37), health facilities that had separate waiting room for youth AOR 9.84, 95%CI (2.14,17.79), and health facilities that established system in place to provide continuous support to staff AOR 2.81, 95%CI(1.25, 6.34) were statistically significant determinants of FoI of the YFS (**Table 6**).

The odds of getting the YFS with fidelity was nearly two times higher among those youth who were served by health care providers who got capacity building training as compared to their counter parts with AOR 1.93, 95% CI(1.12, 3.48). The odds of getting the YFS with fidelity was almost three times higher among those youth who were served by health care providers who got supportive supervision as compared to those youth who had not got supportive supervision with AOR 2.85, 95% CI (1.99, 6.37). The odds of getting the YFS with fidelity was nearly ten times more among those youth who were served from health facilities that had separate waiting room for youth AOR 9.84, 95%CI (2.14,17.79), as compared to those youth who were served from health facilities that had no separate waiting room. The odds of getting the YFS with fidelity was nearly three times higher among those youth who were served from health facilities that already established system in place to provide continuous support to the YFS staff as compared to those who were not with AOR 2.81, 95%CI(1.25, 6.34) (**Table 6**).

## Discussion

The findings of the study showed that level of implementation fidelity remains low; both individual level and contextual level determinants affect the implementation fidelity of YFS. The analysis indicated that FoI of YFS among individual youth depends on the joint effect of individual, health care provider and facility characteristics.

At the individual level variables like youth who had high overall involvement in the provision of YFS, youth who know the presence of trained peer educator in the area and youth who have been involved as a peer educator in YFS were found to be with the main determinants of FoI of YFS. At the contextual level, variables related to YFS providers and YFS program related characteristics were found to be much more relevant for FoI of YFS. A strong facility level determinant for FoI of YFS was related to the provision of capacity building/training to the YFS health care providers.

According to the intra-class correlation results, the contribution of unobserved health facility level characteristics was 16.4%. In all the three intercept-only models, the contributions were significant and indicated that determining association without the control of variables at different levels would give a misleading result. This was also observed during analysis where many of the significant associations disappeared when the effect of clustering by health center was controlled. Previous studies based on a similar analysis showed consistent findings [45].

The first individual level variable which is a strong determinant factor for FoI of YFS was related to youth over all involvement in the YFS provision. The odd of getting the YFS with fidelity is 1.4 times higher among those youth who had high overall involvement in the provision of YFS as compared to those who had not. This finding is supported by a theory developed by Christopher Carroll et al., they verified that as participants involved more in the provision of an intervention the possibility of getting the intervention with fidelity will increase [13]. In addition, the National Adolescent and Youth health strategy also documented as establishing supporting and facilitating youth engagement and ownership of health programs like YFS is an enabling condition to deliver the YFS with high fidelity [3]. Involvement of the youth in the day-to-day planning and running of activities, including monitoring of services ensures the services are of good quality and acceptable to the youth. When the youth perceive

the services to be acceptable, they are most likely to refer the said services to their colleagues and peers [4]. The possible explanation could be as youth are involved more in the provision of interventions like YFS, they have already belonged to the YFS providers and they may ask any services they demand, they may have a better communication, better knowledge on the available services and hence better chance of getting the intervention with fidelity [13]. This implied that involving youth in the YFS intervention is crucial in order to increase the fidelity of implementation of the intervention.

The second individual level factor that affects the implementation fidelity of YFS was related to youth knowledge on the presence of trained peer educator in the area. The odds of getting the YFS with fidelity is 1.6 times higher among those youth who know any peer educator there trained in YFS as compared to those who did not know. This finding is in agreement with a study conducted in Kenya [46], where the study showed that over all youth knowledge on SRH as the main enabler to YFS uptake by youth. In addition, the finding is supported by another study conducted in Myanmar [47], where youth who had better knowledge on SRH including the presence of trained youth in YFS has increased the uptake of YFS. The possible justification could be those youth who already know trained peer educators in the area can have a better chance to discuss with the trained peer on how to communicate with the YFS provider and hence get the intervention with fidelity. Besides, those youth who know peer educators may have a possibility to go to the health facility accompanied with those peer educators so that the peer educator can facilitate the provision of the YFS intervention so that they can get the YFS with high fidelity. Furthermore, as adolescents preferred peer educators as a source of sexual and reproductive information since they considered them knowledgeable and trustworthy [46, 47].

The third individual level variable that was found to be a strong determinant factor for FoI of YFS was related to presence of youth involvement as a peer educator in YFS. The odd of getting the YFS with fidelity is one and half times higher among youth who have been involved as a peer educator in YFS as compared to those youth who were not involved. This finding is supported by the general theoretical frame work developed by Christopher Carroll et al. [13] where participants involvement including youth is mentioned as the main individual level variable that affects the implementation fidelity of interventions. Similarly, this finding is supported by a study conducted in Awabel district, where in the study youth who were participated a peer educators had a higher chance of SRH service utilization [48]. The possible justification could be the more enthusiastic participants are about an intervention, the most likely they get the intervention with a better fidelity [13]. In addition, as young people engaged in SRH peer education; they would have a better understanding and their need for the service and getting the service with high fidelity might increase too [13].

The odds of getting the YFS with fidelity is nearly two times higher among those youth who were served by health care providers who had got capacity building training as compared to their counter parts. This result is supported by the finding of a review article, where capacity building trainings are critical for ensuring retention of the YFS providers' knowledge and skills up to date and hence help them provide the YFS with higher fidelity [49]. Besides this finding is in line with a study conducted in New Mexico and Bahaman where provision of capacity building training to intervention providers was identified as a factor that increased the implementation fidelity of evidence-based practices for integrated treatment in behavioral health agencies [50, 51]. The possible explanation could be providing capacity building training to the intervention providers is one of the possible motivation factor that could help intervention providers develop more confidence and hence provide the intervention with a higher fidelity [22]. In addition, those providers who get training can have a better knowledge and technical

skills on the provision of a specific intervention with a higher fidelity, since they can easily adhere to the intervention protocol [22].

Another strong facility level determinant for FoI of YFS was related to presence of supportive supervision to YFS providers'. The odds of getting the YFS with high implementation fidelity is almost three times higher among those youth who were served by health care providers who have got supportive supervision as compared to those youth who had not. This finding is in agreement with a review research and a study conducted in Addis Ababa that showed program implementation fidelity is clearly predicted by the level of supportive supervision provided to the organizational staff [12, 52]. The possible justification could be as program implementers get technical assistance including the training of program facilitators and program administrators, program evaluation and feedback, program monitoring, and coaching the providers will get more skills; hence the possibility of providing the intervention with fidelity increases [53].

The odds of getting the YFS with fidelity is nearly ten times more among those youth served from health facilities that had a separate waiting room for youth as compared to those youth served from health facilities that had no separate waiting room. This finding is supported by a study conducted in USA where those youth who were served by health facilities that had a separate waiting room showed a better care on retention among HIV-infected youth [54]. Besides, this finding is in line with the WHO quality assessment standard [36] that stated as the presence of separate waiting room for the YFS will enhance the intervention delivery with a better fidelity. The possible explanation could be, usually waiting rooms for youth are equipped with many educational materials like visual and audio, leaf lets, TV, posters and even peer educators there. Hence, youth in the waiting area have a better information on the YFS services available, how to approach the YFS provider and the like [36]. In addition, separate waiting rooms allow youth to be kept from visual and auditory privacy from adult clients and that may increase youth confidence on seeking the YFS and hence all the above may contribute to get the YFS with fidelity [36, 54].

The last strong facility level determinant for FoI of YFS was related to the presence of already established system to provide continuous support to staff who works on YFS. The odds of getting the YFS with fidelity was nearly three times higher among those youth who were served from health facilities that already had established system in place to provide continuous support to staff who work with young clients as compared to those who were not. This finding is similar with a study conducted in New Mexico, where the presence of continuous organizational support to the staff has an influence to the intervention to be implemented with a higher fidelity [50]. Besides, this finding is in line with a study conducted in USA that was intended to evaluate the efforts to increase implementation of evidence-based clinical practices to improve adolescent-friendly reproductive health services [55], where by support from health center leadership, communication between leadership and staff, were reported as factors that facilitated the implementation of new practices [55]. The possible reason could be establishing a system to provide continuous support to staff may motivate and increase commitment of the staff to implement the intervention with a higher fidelity [3, 56].

The results of the study implied that efforts to build systems that apply the policy and principles of implementation fidelity of YFS in the health facilities are crucial. These initiatives will not only benefit youth but the health system overall, as the principles for implementation fidelity of the National Adolescent and Youth Health Strategy are in step with those of YFS to be delivered with a high level of adherence, quality of delivery and youth engagement [3]. Another implication of the study finding is considering high levels of implementation fidelity of YFS is essential to avert the SRH problems among youth, in addition to the YFS scale up. Furthermore, investing in effective interventions (like YFS), is important to improve its

implementation fidelity. There is a need to strengthen the YFS to be delivered with a high level of fidelity to achieve the desired intervention outcomes.

The findings of the study have some policy implications in that, while designing the YFS by considering on both individual and contextual level factors if important to strengthen and provide the YFS with high fidelity if implementation. In addition, the findings also has some practical implications, in that while providing the YFS intervention for youth it is vital to consider the youth involvement in the planning and provision of YFS. Another practical implication is considering contextual level factors like the YFS program level characteristics, the YFS provider and the health facility environment had paramount important.

## Limitations

The study was not triangulated with a qualitative design, which could probably explore in-depth reasons for the low level of fidelity of implementation of YFS. Besides, the measure of Fidelity of implementation of YFS was from youth perspectives, which means, the study did not consider the providers' perspective and direct observation. Hence, some fidelity items that should be filled by the providers were not considered.

## Conclusions

In this study, the level of implementation fidelity remains low. Both individual and contextual level factors are found independent determinants on the FoI of YFS. High overall involvement in the provision of YFS, youths knowledge on the presence of trained peer educator in the area, have been involved as a peer educator in YFS, health care providers who had got capacity building training, health care providers who have got supportive supervision, health facilities that have a separate waiting area for youth clients and health facilities that already established system in place to provide continuous support to staff who work on YFS were predictors of FoI of YFS. Therefore, policy makers, partners, planners, managers and YFS providers could consider both individual and contextual level factors to improve the implementation fidelity of YFS.

## Acknowledgments

The authors acknowledged Central Gondar Zone and Gondar City administration Health Department officials for their support. In addition, administrators of each district health offices and head of the health centers, the data collectors, the supervisors, the YFS providers and the respondents are also acknowledged.

## Author Contributions

**Conceptualization:** Yohannes Ayanaw Habitu, Gashaw Andargie Biks, Abebaw Gebeyehu Worku, Kassahun Alemu Gelaye.

**Data curation:** Yohannes Ayanaw Habitu.

**Formal analysis:** Yohannes Ayanaw Habitu.

**Funding acquisition:** Yohannes Ayanaw Habitu.

**Investigation:** Yohannes Ayanaw Habitu.

**Methodology:** Yohannes Ayanaw Habitu, Gashaw Andargie Biks, Abebaw Gebeyehu Worku, Kassahun Alemu Gelaye.

**Project administration:** Yohannes Ayanaw Habitu.

**Resources:** Yohannes Ayanaw Habitu.

**Software:** Yohannes Ayanaw Habitu.

**Supervision:** Gashaw Andargie Biks, Abebaw Gebeyehu Worku, Kassahun Alemu Gelaye.

**Validation:** Yohannes Ayanaw Habitu, Gashaw Andargie Biks, Abebaw Gebeyehu Worku, Kassahun Alemu Gelaye.

**Visualization:** Yohannes Ayanaw Habitu.

**Writing – original draft:** Yohannes Ayanaw Habitu.

**Writing – review & editing:** Yohannes Ayanaw Habitu, Gashaw Andargie Biks, Abebaw Gebeyehu Worku, Kassahun Alemu Gelaye.

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
