## [Decision Letter · Decision Letter 0]

12 Aug 2021

PONE-D-21-19080

Individual and Contextual Factors affect the Implementation Fidelity of Youth-Friendly Services, northwest Ethiopia: a Multilevel Analysis

PLOS ONE

Dear Dr. Yohannes Ayanaw Habitu,

Thank you for submitting your manuscript to PLOS ONE. After careful consideration, we feel that it has merit but does not fully meet PLOS ONE’s publication criteria as it currently stands. Therefore, we invite you to submit a revised version of the manuscript that addresses the points raised during the review process.

We look forward to receiving your revised manuscript.

Kind regards,

Limakatso Lebina, MBChB

Academic Editor

PLOS ONE

Journal Requirements:

This research was conducted as an academic research contribution. The University of Gondar covered the costs for the data collection procedures. Otherwise, the authors received no specific funding for this work.

Additional Editor Comments:

Thank you for submitting this interesting assessment of fidelity on the implementation of youth services. The major comments on this manuscript are on the method section as indicated by the two reviewers. It is great that you utilised a tool that has been recommended by WHO to assessment tool. However, in the analysis and interoperation of the results you did not follow those guidelines (according to ref 36) - WHO: Quality Assessment Guidebook. A guide to assessing health services for adolescent clients. 2009.

Please explain in the methods section why you did not follow these guidelines on data analysis and interpretation. For example you utilised 60% as a good level of fidelity based on a different article instead of the quality assessment guidebook recommendations.

Reviewers' comments:

Reviewer's Responses to Questions

**Comments to the Author**

1. Is the manuscript technically sound, and do the data support the conclusions?

Reviewer #1: Yes

Reviewer #2: Yes

2. Has the statistical analysis been performed appropriately and rigorously? 

Reviewer #1: Yes

Reviewer #2: I Don't Know

3. Have the authors made all data underlying the findings in their manuscript fully available?

Reviewer #1: No

Reviewer #2: Yes

4. Is the manuscript presented in an intelligible fashion and written in standard English?

Reviewer #1: Yes

Reviewer #2: Yes

5. Review Comments to the Author

Reviewer #1: This is a very interesting and important piece of work so well done for putting in the effort. There is however room for improvement and clarification needed particularly in the methodology section. My concerns are as follows;

Line 87-89: Under your rationale you mention that the undertaking of the study would uncover evidences related to IFidelity of the program. In what way would that happen? what evidences are you talking about?

Line 100-101; Reword the section, make it clear that you were assessing determinants on two levels; individual & program level and be consistent throughout the paper. My understanding is that characteristic of program and provider information fall under "Program level" please be clear.

You mention the assessment of readiness and competencies as well in your methodology, how do these tie in with your objectives. It seems to me you are dealing with too many constructs.

Line 118-120: Please clarify how you got from the 8 Health centres to 11 health centres. This comment is tied in with your sampling procedure. My suggestion is you reword it in a logical sequence so one is clear how you got to the 11 centers.

Calrify also what is "lottery method?" is this the same as random sampling?

You have too many variables, i suggest you categories into the two levels you are assessing and be consistent.

Line 393, this section on determinants of IF sounds like it should go into the analysis section.

Check your tenses and grammar under your results.

The discussion is clear and well written. All the best with your revisions.

Reviewer #2: Review (Manuscript: PONE-D-21-19080)

August 05, 2021

This is a good paper for the field of implementation science. Youth-friendly service programs (YFS) are a fantastic way to link and retain youth in care services. Given the broad implementation of the YFS program in parts of Ethiopia (as noted by the authors), this research is necessary to the field. Overall, the approach to assessing fidelity I also think is valuable. However, there were some points in the manuscript that confused me as a reader. To use the Carroll et al. (2007) study you cited, implementation fidelity is focused on the degree to which an intervention is delivered as intended. In my view, an evaluation of implementation will be focused on the health facility and the providers. This is not to say that clients (i.e., youth, adolescents, etc.) cannot evaluate fidelity, but that their role in measuring the success of the intervention is different.

And this is where I came off confused. You, the authors, are looking at the fidelity of implementation, and your focus appears to be clients, those youth who seek care services from the facilities in Gondar, Ethiopia. And you don’t do a good job explaining why this became your target population. Typically, the clients (i.e., youth) will be used to assess adherence to the intervention. Then, for example, a high adherence becomes an indicator of the success of implementation.

See below for more specific comments about different sections of the manuscript.

Introduction

Line 78, I think you meant to write fidelity of implementation and not “fidelity f implementation”

I may be misunderstanding the manuscript, but why is the question of fidelity being posed to youth? Who is the target population? Fidelity would seem to be an issue for the program implementers and providers. The providers and program implementation team would be the ones to assess whether providers delivered YFS as it was designed.

The question for the youth would be one of adherence. The success of YFS for youth would be whether they came for services and continued to come for services for the length of the program can be given a better assessment of w

Methods

The section on instrument development is a bit long, with considerable repetition. For example, you mention that aspects of the final instrument were adapted from the WHO standard on Line 163 and 216. Consider shortening this section, given that you are interested in one component of the instrument, which is fidelity. It would help if you also considered writing a manuscript on the instrument development or the study protocol. The development of the instrument is itself an interesting process with worthwhile contributions to the literature.

What are some examples of questions used in the fidelity component of the instrument?

It is not clear why the study (or the part of the study reported in this manuscript) has both youth and providers as the study population. Specifically, was the instrument given to providers the same as the instrument given to the youth participants? Why is that? Providers would have a different perspective on issues around fidelity; I would expect you to look to capture those differences.

And again, if it is the fidelity of implementation, why do you need the perspective of the youth. I don’t imagine that the youth, who are clients of these health care facilities, would have a significant say in how a program is implemented. Or at least they would have a lesser say compared to providers.

How was fidelity understood and defined by the authors? Reading the manuscript, I feel you assume your readers have the same understanding of fidelity as you do. For example, in the methods the items used to capture fidelity, what domains do they cover. You mention that it is based on an existing instrument used by the WHO. But what aspects/constructs of fidelity does it cover?

Results

You report youth and health care providers together in Table 01; should that not be separated? The factors that impact providers when it comes to the question of fidelity I imagine, are distinct.

Table 02 and Table 03 cover issues that are not discussed in the methods section. That is level of youth engagement and facility-level factors are not evident in the methods. For table 02, in particular, what is meant by the level of engagement?

Table 02 could be made more clear if you categorized the engagement levels. When I reference Table 05, it is clear that the levels of engagement form part of the independent variables in your multi-level model. So, for example, if model II was adjusted for individual-level variables, then have Table 02 show this so that your readers can better connect Table 02 and Table 05.

Consider shortening the Items/ variable names in Tables 02, 03, and 05; it makes the tables clunky and challenging to read.

Discussion

Is there a reason you reference implementation in education and not the implementation in health care to support your point here? It is not wrong, but I think your argument would be stronger if you found implementation studies in the health sector and the sexual health field specifically to support notions of fidelity of implementation (Lines 512 - 514).

So, what are the practice implications of your study findings? YFS programs are demonstrated in the literature to be effective and successful ( I think several clinical studies bear this out). The question then is the state of fidelity of implementation in Gandor, Ethiopia, and more importantly, what would the iteration and scale-up of YFS programs in Ethiopia look like, given your analysis data? I think this should be a significant focus of your discussion. Ethiopia has already begun implementing YFS programs, so what will need to change to enhance fidelity in light of the results?

Limitation

You may wish to say a bit more regarding the limitations of the study. What do the survey items on fidelity leave out when it comes to understanding implementation?

6. PLOS authors have the option to publish the peer review history of their article (what does this mean?). If published, this will include your full peer review and any attached files.

Reviewer #1: No

Reviewer #2: No

---

## [Author Response · Author response to Decision Letter 0]

22 Sep 2021

Point by point response to Editor’s comments

Thanks a lot dear Editor, for the efforts and support you made to improve our manuscript. In the table below, we write the Editor comment in column 1, the Author responses in column 2 and page/line numbers in column 3 to show the changes we made in the revised clean manuscript. 

Editor’s Comments Authors’ responses Page and line #, R(Implies Authors' responses), C (Implied Editor's or reviewers' Comment )

C: Thank you for submitting your manuscript to PLOS ONE. After careful consideration, we feel that it has merit but does not fully meet PLOS ONE’s publication criteria as it currently stands. Therefore, we invite you to submit a revised version of the manuscript that addresses the points raised during the review process. 

R: We thank the Editor for providing the chance to review and submit the revised manuscript. We have tried to address almost all of the points raised by you and the two reviewers. -

R: Thanks again. We have cheeked the manuscript to meet the PLOS ONE’s style requirements including the file naming. -

R: Dear editor, we ask an excuse for the information mismatch provided in the ‘Funding Information’ and ‘Financial Disclosure’ sections on the grant information.

Now we have included the correct grant numbers for the awards we received from the University of Gondar in the Funding Information Section.

“The University of Gondar provided the grant with the grant number:

 No: Res/Com/Serv/Vice/Pres/05/1203/2011 -

R: This research was conducted as an academic research contribution. The University of Gondar covered the costs for the data collection procedures. Otherwise, the authors received no specific funding for this work.

R: Thanks again dear editor for asking further clarity.

The funder/The University of Gondar/ has no role in the study design, data collection and analysis, decision to publish or preparation of the manuscript.

We have incorporated this statement as “The funders had no role in study design, data collection and analysis, decision to publish, or preparation of the manuscript”, in the cover letter. -

Additional Editor Comments:

C: Thank you for submitting this interesting assessment of fidelity on the implementation of youth services. The major comments on this manuscript are on the method section as indicated by the two reviewers. It is great that you utilised a tool that has been recommended by WHO to assessment tool. However, in the analysis and interoperation of the results you did not follow those guidelines (according to ref 36) - WHO: Quality Assessment Guidebook. A guide to assessing health services for adolescent clients. 2009.

Please explain in the methods section why you did not follow these guidelines on data analysis and interpretation. For example you utilised 60% as a good level of fidelity based on a different article instead of the quality assessment guidebook recommendations. 

R: Thanks again for your appreciation of the manuscript.

The major comments provided by the two reviewers on the methods section of the manuscript are addressed in the revised version.

Dear editor, you are right. We utilized the WHO quality assessment tool to generate most of the questions used to measure fidelity. However, we did not use the WHO cutoff value (>=75%) in the data analysis and interpretation sections. The reason why we did not use the WHO cutoff value was, while we were using the WHO cutoff value the proportion of youth who get the YFS with higher fidelity became very small(4.7%), and it was very difficult to run/fit/ the multilevel modeling due to small proportion of youth get the YFS with good fidelity. Hence, we have to review the available evidence and use 60% as a cut off value.

We have included the reasons why we used 60% cutoff value instead of using the WHO cutoff value in the methods section of the revised manuscript as:

“In this study, the WHO cutoff value (>=75%, ≥243.75/325) was not used in the data analysis and interpretation sections to declare good fidelity. The reason why we did not used the WHO cutoff value was, while we were using the WHO cutoff value, the proportion of youth who get the YFS with higher fidelity became very small, 48 (4.7%). Which made the data analysis very difficult and therefore running/fitting/ the multilevel modeling using such small proportions was impossible. Hence, we reviewed the available evidence and used 60% as a cut off value.” Page 13 Line 264-270

Point by point response to Reviewer one

R: First of all, we want to appreciate reviewer #1 for evaluating our manuscript and providing these constructive comments that potentially improve the revised manuscript to be accepted for publication. We have tried to incorporate almost all the comments and suggestions made by the reviewer. The reviewer comments (column 1), the author responses (column 2) and the place where the changes we made in the revised manuscript are described in column 3 of the table below. 

Reviewer # 1 Comments Author responses Page/Line #

C: This is a very interesting and important piece of work so well done for putting in the effort. There is however room for improvement and clarification needed particularly in the methodology section. My concerns are as follows; 

R: Thanks a lot, dear reviewer for providing your immense appreciation for our work and the efforts we made. 

Again we recognized your requests for some clarifications and concerns, which will help for the improvement of the manuscript. We have tried to address all of your concerns and comments you raised especially in the methodology section. -

C: Line 87-89: Under your rationale you mention that the undertaking of the study would uncover evidences related to IFidelity of the program. In what way would that happen? what evidences are you talking about?

R: Thanks for requesting further clarification for the evidences that this study tried to uncover. 

The determinants of Fidelity of Implementation (FoI) of interventions (like YFS) are usually not clear unless otherwise investigations are made. There is no previous study conducted to uncover or show the determinants of FoI of YFS. Other evidences from similar interventions showed the determinants of FoI are related to the program or to the clients’ side. In this case, we used the term ‘the evidences’ to mean ‘the determinants’. Hence, this study was conducted to verify the determinants of FoI of YFS that could arise from (related to) the program side or from the user side. To verify those determinants we employed an advanced statistical modeling named as the multi-level modeling approach that helped us to clearly verify the determinants by considering program and individual levels. 

To make it clear for the reader we amended the statement as: “Investigating the determinants of FoI of YFS using the multilevel perspective is very important, in that, it will uncover evidences related to the determinants of implementation fidelity of YFS.”; and included in the revised manuscript. Page 5 Line 88-90

C: Line 100-101; Reword the section, make it clear that you were assessing determinants on two levels; individual & program level and be consistent throughout the paper. My understanding is that characteristic of program and provider information fall under "Program level" please be clear. 

R: We appreciated the reviewer for providing such constructive suggestions. 

You are right dear reviewer! Program and provider-level characteristics fall under the program-level characteristics. Based on your suggestions we made changes in the whole manuscript. We put program and provider information into program level determinants. Page 5 Line 100-101

C: You mention the assessment of readiness and competencies as well in your methodology, how do these tie in with your objectives. It seems to me you are dealing with too many constructs.

R: Yes, you are correct dear reviewer. We have assessed the health centers level of readiness to provide the YFS by using the WHO standard checklist. Unfortunately, we did not get any determinant factor that affected the FoI of YFS from the health center level of readiness side. Besides, we also employed a separate checklist (that was also adopted from the WHO standard checklist) which was intended to assess the level of YFS providers’ competency in the provision of YFS. Similarly, we did not get any determinant in this construct. 

The study objective was to assess the determinants of FoI of YFS using the Multilevel perspective by considering many constructs that are described in the paper.

Really we employed many constructs to deal with the potential determinants of FoI of YFS in this study. -

C: Line 118-120: Please clarify how you got from the 8 Health centres to 11 health centres. This comment is tied in with your sampling procedure. My suggestion is you reword it in a logical sequence so one is clear how you got to the 11 centers.

Calrify also what is "lottery method?" is this the same as random sampling? 

R: We ask an apology for the confusion we imposed on the procedure we wrote for the selection of the health centers. The 11 Health centers, which were included in the study, were selected from two areas. The two areas are the Central Gondar Zone administration and the Gondar City administration. Gondar city administration (located at the center of Central Gondar Zone administration) is the capital city of the Central Gondar Zone administration.

The five health centers were selected randomly from the health centers that were found under the Central Gondar Zonal administration and that were implementing the YFS. On the other hand, the 6 health centers were selected from the available health centers that were implementing the YFS in Gondar city administration. 

The sampling procedure is rephrased logically and included in the revised manuscript as:

“In Central Gondar Zone, there are 14 rural districts and one city administration. Out of the 14 rural districts, 5 districts were selected by simple random sampling technique. Then, if there are two or more health centers implementing YFS in each district, 1 health center, was chosen by random sampling. Hence, 5 health centers from the Central Gondar Zone (Amba Giorgis, Maksegnit, Kolladiba, Chuahit and Delgie) were selected and included. On the other hand, from the 8 health centers that were implementing the YFS in Gondar city administration, 6 health centers (Gondar, Azezo, Tseda, Gebriel, Woleka and Maraki) were selected randomly and included in the study. Finally, when we sum up those selected health centers (from the Central Gondar Zone (5 HCs) and Gondar City Administration (6 HCS)), a total of 11 health centers were included and considered as clusters. Within each cluster, the YFS program strength, the health facility readiness and the YFS providers’ competency was assessed.”

R: In this study “lottery method” is the same as “random sampling”. To make it clear for the reader we replaced the phrase “lottery method” with “random sampling” in the revised manuscript. Page 7 Line 140-151, Page 7 Line 143-144

C: You have too many variables, i suggest you categories into the two levels you are assessing and be consistent. 

R: We accepted the reviewer suggestions.

Now we have categorized the variables into two levels and revised the manuscript based on the reviewer suggestions. Page 10 Line 215-218, Page 11 Line 219-234

C: Line 393, this section on determinants of IF sounds like it should go into the analysis section.

R: Dear reviewer, the section you stated on the determinants of IF seam sounds like it should go into the analysis section. However, the issue you raised here is a bit different. The analysis section of the manuscript (in the methods part) described the overall data analysis plan, where the authors of the manuscript plan ahead of the actual data collection period. On the other hand, what we wrote in Line 393 supported by Table 5 (in the results section of the manuscript), included the actual multilevel analysis results, which were described by showing each step with its subsequent findings while we run the Multilevel modeling. We believe that this is the appropriate place to show (describe) the steps and the results of the multilevel analysis in the results section. 

 Most papers published (that employed the multilevel model) also put or described this findings (in the results section) (e.g., Ayal Debie et al., 2020, Negero et al., 2018, Worku et al., 2021., etc… ). -

C: Check your tenses and grammar under your results.

R: Comment accepted. We have cheeked and revised the tenses and grammar under the results section. Page 17 Line 349-357, Page 18 Line 358-367, Page 19 Line 370-378,

Page 20 Line 382-393, Page 22 Line 405-422, Page 23 Line 423-444, Page 24 Line 445-449

C: The discussion is clear and well written. All the best with your revisions.

R: Thanks for the interest and commitment you made to review the whole paper, provide constructive comments and evaluate the whole manuscript.

Again thanks a lot dear reviewer, for the best wishes you forwarded to us! -

Point by point response to Reviewer Two

R: First of all, we want to appreciate reviewer #2 for evaluating our manuscript and providing these constructive comments that potentially improve the revised manuscript and make it to be accepted for publication. We have tried to address most of the comments and suggestions made by the reviewer. The reviewer comments (column 1), the author responses (column 2) and the place where the changes we made in the revised manuscript are described in column 3 of the table below. 

Reviewer #2 Comments Author responses Page/Line #

C: This is a good paper for the field of implementation science. Youth-friendly service programs (YFS) are a fantastic way to link and retain youth in care services. Given the broad implementation of the YFS program in parts of Ethiopia (as noted by the authors), this research is necessary to the field. Overall, the approach to assessing fidelity I also think is valuable. However, there were some points in the manuscript that confused me as a reader. To use the Carroll et al. (2007) study you cited, implementation fidelity is focused on the degree to which an intervention is delivered as intended. In my view, an evaluation of implementation will be focused on the health facility and the providers. This is not to say that clients (i.e., youth, adolescents, etc.) cannot evaluate fidelity, but that their role in measuring the success of the intervention is different.

R: Thanks a lot, dear reviewer for your appreciation of the manuscript. 

Dear reviewer, as you have mentioned implementation fidelity is measured by using different perspectives (study population) ((Fixsen, D. L., Van Dyke, M. K., & Blase, K. A. (2019). Implementation methods and measures. Chapel Hill, NC: Active Implementation Research Network)). 

Some authors use the health facility and providers’ responses to evaluate the implementation fidelity of interventions. Others use direct observation to evaluate the implementation fidelity. Still, some researchers use a camera or audio recording to measure implementation fidelity (Elaine Toomey et al., 2017). Furthermore, many scholars use service users’/clients’ responses to evaluate the implementation fidelity of interventions (Silvia Escribano et al., Spain, 2016, Lisha et al., 2012).

During the design stage of the current study, we (the authors of this manuscript) made arguments to select the best method (study population) to evaluate the implementation fidelity of the YFS. Pieces of evidence showed using the service providers as a source for measuring implementation fidelity showed inflated results (eg., Sarah K. et. al., Wang et al. 2015). As the service providers are part of the intervention/as a provider/, usually they give a high score for each fidelity measure(dimension) and finally, the overall fidelity of implementation score will be inflated and which will not give space for improving the program implementation. Direct observation and camera recordings have many ethical issues and are resource-intensive; hence those methods are very difficult to apply in an Ethiopian (our) setup. After taking so many methodological reviews, we reached at a consensus to use the responses’ of the clients/youths/ to measure the fidelity of implementation fidelity of YFS in the current study. -

C: And this is where I came off confused. You, the authors, are looking at the fidelity of implementation, and your focus appears to be clients, those youth who seek care services from the facilities in Gondar, Ethiopia. And you don’t do a good job explaining why this became your target population. Typically, the clients (i.e., youth) will be used to assess adherence to the intervention. Then, for example, a high adherence becomes an indicator of the success of implementation.

R: We thank the reviewer for asking further clarification. The YFS intervention was designed (by WHO, 2001) to be delivered to youth (15 -24 years old). Ethiopia adopted the YFS intervention from the WHO, and also delivers the YFS for youth aged 15 – 24 years. That is the reason why youth are our target population. 

The YFS is designed to be delivered by health care providers who took pre-service training on the YFS. The checklists designed to evaluate the implementation fidelity of YFS from the youth perspectives are almost equivalent as compared to the checklist designed for the providers. The reason why we collected data from the YFS providers and the health facilities as study participants was to assess the determinants of the fidelity of implementation of YFS at the program level (in addition to the individual level determinants). 

In this study, the fidelity of implementation was developed from three main constructs. These constructs were adherence, quality of YFS delivery, and the youth engagement in the YFS intervention. -

Introduction

C: Line 78, I think you meant to write fidelity of implementation and not “fidelity f implementation”

R: Thanks to the reviewer for the spelling corrections you suggested. We have corrected the spelling mistake and write it as “fidelity of implementation”, in the revised manuscript. Page 4 Line 79

C: I may be misunderstanding the manuscript, but why is the question of fidelity being posed to youth? Who is the target population? Fidelity would seem to be an issue for the program implementers and providers. The providers and program implementation team would be the ones to assess whether providers delivered YFS as it was designed.

R: Dear reviewer, the measure of fidelity of implementation of an intervention like the YFS can be assessed by using different perspectives (respondents). 

The program implementers, the providers and/or the users/clients’ (youth) can be the source of information to evaluate (measure) the implementation fidelity of an interventions (like the YFS). Using either of the study population as a source of information to measure implementation fidelity has its own limitations and strengths. After having a deep discussion with the research team (during the design phase of this study) we have agreed to use youth as a study population (source of information) to evaluate (measure) the fidelity of implementation of the YFS. We provided the interviewer-administered questionnaire for each youth just after they exit the YFS room immediately (after) youth got the YFS.

As stated above, youth are the target population for this study for the measure of the fidelity of implementation of YFS. Moreover, we used the YFS providers and the YFS program (health center) as another study population to assess the determinants of the implementation fidelity of the YFS at a program level. Since the determinants may came (arise) from the YFS program side or from the YFS users’ side (youth).

Finally, the providers and program implementation team should not be the ones to assess whether providers delivered the YFS as it was designed. Because, the providers and the program implementation team are part of the implementation and the results will probably be inflated /maybe near to 100 %/ due to bias (Hawthorne effect). The Hawthorne effect is a type of bias that referring to the tendency of some people (here the YFS providers) to work harder and perform better when they are participants in an experiment (Adeoti et al., The fidelity of implementation of recommended care for children with malaria by community health workers in Nigeria, 2020(94% adherence level was found in this study)) & ((Asgary-Eden, V., & Lee, C. M. (2011). So now we've picked an evidence-based program, what's next? Perspectives of service providers and administrators. Professional Psychology: Research and Practice, 42(2), 169–175. (In this study the average adherence rate reported by the service providers who used the program was 85.9%)).). Therefore, to minimize or avoid the effect of Hawthorne effect, the evaluators should be independent researchers like us. We the current research team are scholars/academicians and researchers/ working at the University of Gondar and have a deep understanding of the YFS intervention. We are not participating as a provider or as a program implementer in the YFS intervention currently. -

C: The question for the youth would be one of adherence. The success of YFS for youth would be whether they came for services and continued to come for services for the length of the program can be given a better assessment of w 

R: Regarding the items we included, questions that were designed to assess adherence were incorporated. In addition, questions designed to assess the quality of delivery and the participant engagements were also included. Page 8 Line 166-173, Page 9 Line 174-176

Methods

C: The section on instrument development is a bit long, with considerable repetition. For example, you mention that aspects of the final instrument were adapted from the WHO standard on Line 163 and 216. Consider shortening this section, given that you are interested in one component of the instrument, which is fidelity. It would help if you also considered writing a manuscript on the instrument development or the study protocol. The development of the instrument is itself an interesting process with worthwhile contributions to the literature. 

R: Thanks a lot, dear reviewer for providing this constructive comment. 

We also believe that the instrument development part was long. Now we made a revision on it and tried to make it short and precise. Based on your recommendations, we have planned to write another manuscript regarding the instrument development or the study protocol. Page 8 Line 156-165, Page 8 Line 171-173, Page 9 Line 174-188

C: What are some examples of questions used in the fidelity component of the instrument?

R: Thanks again dear reviewer for requesting some examples of questions used for the fidelity component of the instrument. Based on your request below are some of the questions used to assess the fidelity component in the three domains are described and included in the revised manuscript as:

“Some of the questions used in the fidelity measure based on the three domains (adherence, quality of delivery and participant responsiveness) are described below. Questions used in the adherence domain were: Confidentiality of the service was assured for you, the provider was respectful to you and the provider explained to you all the range of available YFS there. Questions used in the quality of delivery domain were: The hours and day that you came to the facility were convenient for you, You were very clear of the information given by the provider, You are welcomed and get the YFS without an appointment, and the provider encouraged you to ask any questions. Questions used in the participant responsiveness domain were: You were involved as a peer educator in YFS, you were involved in contributing to decisions about how health services should be delivered to youth clients, and you were involved in YFS service design and delivery.” Page 9 Line 189-196, Page 10 Line 197-198

C: It is not clear why the study (or the part of the study reported in this manuscript) has both youth and providers as the study population. Specifically, was the instrument given to providers the same as the instrument given to the youth participants? Why is that? Providers would have a different perspective on issues around fidelity; I would expect you to look to capture those differences.

R: Dear reviewer, we are sorry for the confusion we made on this issue. Let us explain it more. Both youth and health care providers are the study population for this specific research. The information from the youth was used to assess the fidelity of implementation of YFS, while the information from the health care providers (YFS providers) was used to assess the providers’ level of competency in the provision of the YFS. Besides, information from the health care providers was used to assess the level of health facility readiness in the provision of the YFS. The fidelity instrument that was filled by youth is different from the instrument prepared for the health care providers. -

C: And again, if it is the fidelity of implementation, why do you need the perspective of the youth. I don’t imagine that the youth, who are clients of these health care facilities, would have a significant say in how a program is implemented. Or at least they would have a lesser say compared to providers. 

R:Thanks again dear reviewer. Scholars used different approaches to measure the fidelity of implementation of interventions like the YFS. Some use providers as a study population. Others use clients/here youth/ as a study population. Still, others use the combination from providers and clients by preparing to provide similar questionnaires for the provider and the clients. Using this combined method is resource-intensive and findings showed the fidelity measure that was from the providers was inflated or higher compared to the fidelity measure from the clients’ side (Asgary-Eden, V., & Lee, C. M. (2011). So now we've picked an evidence-based program, what's next? Perspectives of service providers and administrators. Professional Psychology: Research and Practice, 42(2), 169–175. (In this study the average adherence rate reported by the service providers who used the program was 85.9%)). Still, some other scholars use audio and video recording as well as direct observation as a source to measure the fidelity of the implementation of interventions. The final method has a lot of ethical issues to employ in our setup.

We the authors of this manuscript had a lot of arguments during the design stage of this research and agreed and conducted to use the youth as a source of information to the fidelity measure. -

C: How was fidelity understood and defined by the authors? Reading the manuscript, I feel you assume your readers have the same understanding of fidelity as you do. For example, in the methods the items used to capture fidelity, what domains do they cover. You mention that it is based on an existing instrument used by the WHO. But what aspects/constructs of fidelity does it cover?

R: Thanks, dear reviewer for providing these constructive comments. We accepted the comments and included the definition of fidelity in this specific research as:

“Fidelity of implementation is defined as the extent to which youth get the YFS intervention as compared to the original YFS program protocol based on the three domains called adherence, quality of service delivery and participant responsiveness). ”

 In addition, we have incorporated the different domains/aspects/constructs of fidelity covered in this study by citing additional references that we have based on. The included concepts are written as: 

“To Measure, the implementation fidelity of services like YFS is conceptually developed from three major constructs called adherence, quality of service delivery and participant responsiveness [37, 38]. Hence, it is vital to quantify the three main constructs that are intended to measure the fidelity of YFS [37, 38]. To measure the overall fidelity of YFS, the three constructs used to measure the fidelity of YFS (adherence, quality of YFS delivery and participant responsiveness) were quantified separately”. Page 11 Line 236-238, Page 9

Line 189-196

Results

C: You report youth and health care providers together in Table 01; should that not be separated? The factors that impact providers when it comes to the question of fidelity I imagine, are distinct.

R: You are right dear reviewer and your comments are accepted. We reported the youth and health care provider’s data together in table one. 

Now based on your comments we separately reported the youth (Table 1) and health care providers (Table 2) responses separately (in separate tables).

In the revised manuscript the number of tables became 6. Re-arrangements and renaming of all the tables were made. Page 17 Line 350-357, Page 18 Line 358-367

C: Table 02 and Table 03 cover issues that are not discussed in the methods section. That is level of youth engagement and facility-level factors are not evident in the methods. For table 02, in particular, what is meant by the level of engagement?

R: Comment appreciated. 

Now we have included brief descriptions of the level of youth engagement or participant responsiveness (reported in Table 2) and facility-level factors (reported in table 3) in the methods section.

“In this study level of youth engagement is defined as the participation of youth in the YFS intervention in aspects like participation in the YFS design, planning and delivery, participation in the YFS as a peer educator/counselor, Involvement in the YFS on decisions about how health services should be delivered to youth clients and the like. ”

“Facility level factors are those characteristics like if the health facility has signal listing for all the YFS available or not, if the health facility had a separate discreet entrance for youth to ensure youth privacy or not, if the health facility offered YFS for free or at rates affordable to youth if the health facility has adequate fund allocated for YFS, if the health facility had clear, written guidelines or standard operating procedures exist for YFS and the like.” 

Page 10 Line 199-202, Page 8 Line 166-170

C: Table 02 could be made more clear if you categorized the engagement levels. When I reference Table 05, it is clear that the levels of engagement form part of the independent variables in your multi-level model. So, for example, if model II was adjusted for individual-level variables, then have Table 02 show this so that your readers can better connect Table 02 and Table 05.

R: We accept the comment and revised accordingly.

 In table 02(Now Table 3 in the revised manuscript), those respondents who said ‘yes’ represent those youth who had the level of engagement in the YFS provision. In the original instrument, they responded to the scale as either Agree or Strongly Agree are categorized as having engaged in the YFS provision.

Those respondents who said ‘no’ represent those respondents who had no level of engagement in the YFS provision. In the original instrument, they responded to the scale as either Neutral or Disagree or Strongly disagree are categorized as having no level of engagement. 

To make clear for the reader we have added sub-headings in Table 5(Now Table 6 in the revised manuscript), that clearly demarcated those individual and program-level determinants separately. Page 19 Line 377-378, Page 24 Line 447-449

C: Consider shortening the Items/ variable names in Tables 02, 03, and 05; it makes the tables clunky and challenging to read. 

R: Comment accepted. 

We have shortened the variable names that were described in Tables 02(Now Table 3), 03(Now Table 4) and 05 (Now Table 6). Page 19 Line 377-378, Page 21 Line 402-403,

Page 24 Line 447-449

Discussion

C: Is there a reason you reference implementation in education and not the implementation in health care to support your point here? It is not wrong, but I think your argument would be stronger if you found implementation studies in the health sector and the sexual health field specifically to support notions of fidelity of implementation (Lines 512 - 514).

R: We accept the comment. Dear reviewer the reason why we compared the current finding with the cited implementation research conducted in education was due to the gaps in existing evidence conducted in the health care set-up, specifically in sexual and reproductive health. 

Now after making an extensive literature review we have got one related review article and made the revisions accordingly.

The revisions are included in the revised manuscript as:

“This result is supported by the finding of a review article, where capacity building training is critical for ensuring retention of the YFS providers’ knowledge and skills up to date and hence help them provide the YFS with higher fidelity [49].” Page 27 Line 514-516

C: So, what are the practice implications of your study findings? YFS programs are demonstrated in the literature to be effective and successful ( I think several clinical studies bear this out). The question then is the state of fidelity of implementation in Gandor, Ethiopia, and more importantly, what would the iteration and scale-up of YFS programs in Ethiopia look like, given your analysis data? I think this should be a significant focus of your discussion. Ethiopia has already begun implementing YFS programs, so what will need to change to enhance fidelity in light of the results?

R: Thanks for the comment. Now we have added some practice implications of the YFS based on our findings and included in the revised manuscript as:

“The results of the study implied that efforts to build systems that apply the policy and principles of implementation fidelity of YFS in the health facilities are crucial. These initiatives will not only benefit youth but the health system overall, as the principles for implementation fidelity of the National Adolescent and Youth Health Strategy are in step with those of YFS to be delivered with a high level of adherence, quality of delivery and youth engagement [3]. Another implication of the study finding is considering high levels of implementation fidelity of YFS is essential to avert the SRH problems among youth, in addition to the YFS scale up. Furthermore, investing in effective interventions (like YFS), is important to improve its implementation fidelity. There is a need to strengthen the YFS to be delivered with a high level of fidelity to achieve the desired intervention outcomes” 

Page 29 Line 562-564, Page 30 Line 565-571

Limitation

C: You may wish to say a bit more regarding the limitations of the study. What do the survey items on fidelity leave out when it comes to understanding implementation? 

R: We accepted the comment. Now we added some limitations of the study.

“The measure of Fidelity of implementation of YFS was from youth perspectives, which means, the study did not consider the providers’ perspective and direct observation. Hence, some fidelity items that should be filled by the providers were not considered ” Page 30

Line 581-584

---

## [Editor Report · Decision Letter 1]

26 Jan 2022

Individual and Contextual Factors affect the Implementation Fidelity of Youth-Friendly Services, northwest Ethiopia: a Multilevel Analysis

PONE-D-21-19080R1

Dear Dr. Yohannes Ayanaw Habitu

We’re pleased to inform you that your manuscript has been judged scientifically suitable for publication and will be formally accepted for publication once it meets all outstanding technical requirements.

Kind regards,

Limakatso Lebina, MBChB, Ph.D.

Academic Editor

PLOS ONE
---

## [Editor Report · Acceptance letter]

31 Jan 2022

PONE-D-21-19080R1 

Individual and Contextual Factors affect the Implementation Fidelity of Youth-Friendly Services, northwest Ethiopia: a Multilevel Analysis 

Dear Dr. Habitu:

I'm pleased to inform you that your manuscript has been deemed suitable for publication in PLOS ONE. Congratulations! Your manuscript is now with our production department. 

Kind regards, 

on behalf of

Dr. Limakatso Lebina 

Academic Editor

PLOS ONE